# Provable Gradient Variance Guarantees for Black-Box Variational Inference

**Justin Domke**
College of Information and Computer Sciences
University of Massachusetts Amherst
domke@cs.umass.edu

## Abstract

Recent variational inference methods use stochastic gradient estimators whose variance is not well understood. Theoretical guarantees for these estimators are important to understand when these methods will or will not work. This paper gives bounds for the common "reparameterization" estimators when the target is smooth and the variational family is a location-scale distribution. These bounds are unimprovable and thus provide the best possible guarantees under the stated assumptions.

## 1 Introduction

Take a distribution $p(\boldsymbol{z}, \boldsymbol{x})$ representing relationships between data $\boldsymbol{x}$ and latent variables $\boldsymbol{z}$. After observing $\boldsymbol{x}$, one might wish to approximate the marginal probability $p(\boldsymbol{x})$ or the posterior $p(\boldsymbol{z}|\boldsymbol{x})$. Variational inference (VI) is based on the simple observation that for any distribution $q(\boldsymbol{z})$,

$$\log p(\boldsymbol{x}) = \underbrace{\mathop{\mathbb{E}}_{\mathsf{z} \sim q} \log \frac{p(\mathsf{z}, x)}{q(\mathsf{z})}}_{\text{ELBO}(q)} + KL\left(q(\mathsf{z}) \| p(\mathsf{z}|\boldsymbol{x})\right). \tag{1}$$

VI algorithms typically choose an approximating family $q_{\boldsymbol{w}}$ and maximize $\text{ELBO}(q_{\boldsymbol{w}})$ over $\boldsymbol{w}$. Since $\log p(\boldsymbol{x})$ is fixed, this simultaneously tightens a lower-bound on $\log p(\boldsymbol{x})$ and minimizes the divergence from $q_{\boldsymbol{w}}(\boldsymbol{z})$ to the posterior $p(\boldsymbol{z}|\boldsymbol{x})$.

Traditional VI algorithms suppose $p$ and $q_{\boldsymbol{w}}$ are simple enough for certain expectations to have closed forms, leading to deterministic coordinate-ascent type algorithms [6, 1, 20]. Recent work has turned towards stochastic optimization. There are two motivations for this. First, stochastic data subsampling can give computational savings [7]. Second, more complex distributions can be addressed if $p$ is treated as a "black box", with no expectations available [9, 15, 19]. In both cases, one can still estimate a *stochastic* gradient of the ELBO [17] and thus use stochastic gradient optimization. It is possible to address very complex and large-scale problems using this strategy [10].

These improvements in scale and generality come at a cost: Stochastic optimization is typically less reliable than deterministic coordinate ascent. Convergence is often a challenge, and methods typically use heuristics for parameters like step-sizes. Failures do frequently occur in practice [22, 11, 4].

To help understand when black-box VI can be expected to work, this paper investigates the variance of gradient estimates. This is a major issue in practice, and many ideas have been proposed to attempt to reduce the variance [8, 5, 12, 2, 18, 13, 14, 16]. Despite all this, few rigorous guarantees on the variance of gradient estimators seem to be known (Sec. 5.1).

## 1.1 Contributions

This paper studies "reparameterization" (RP) or "path" based gradient estimators when $q_{\boldsymbol{w}}$ is in a multivariate location-scale family. We decompose $\mathrm{ELBO}(q_{\boldsymbol{w}}) = l(\boldsymbol{w}) + h(\boldsymbol{w})$ where $h(\boldsymbol{w})$ is the entropy of $q_{\boldsymbol{w}}$ (known in closed-form) and $l(\boldsymbol{w}) = \mathbb{E}_{\mathsf{z} \sim q_{\boldsymbol{w}}} \log p(\boldsymbol{z}, \boldsymbol{x})$. The key assumption is that $\log p(\boldsymbol{z}, \boldsymbol{x})$ is (Lipschitz) *smooth* as a function of $\boldsymbol{z}$, meaning that $\nabla_{\boldsymbol{z}} \log p(\boldsymbol{z}, \boldsymbol{x})$ can't change too quickly as $\boldsymbol{z}$ changes. Formally $f(\boldsymbol{z})$ is $M$-smooth if $\|\nabla f(\boldsymbol{z}) - \nabla f(\boldsymbol{z}')\|_2 \leq M \|\boldsymbol{z} - \boldsymbol{z}'\|_2$.

**Bound for smooth target distributions:** If g is the RP gradient estimator of $\nabla l(\boldsymbol{w})$ and $\log p$ is $M$-smooth, then $\mathbb{E} \|\mathsf{g}\|^2$ is bounded by a quadratic function of $\boldsymbol{w}$ (Thm. 3). With a small relaxation, this is $\mathbb{E} \|\mathsf{g}\|^2 \leq a M^2 \|\boldsymbol{w} - \bar{\boldsymbol{w}}\|^2$ (Eq. (3)) where $\bar{\boldsymbol{w}}$ are fixed parameters and $a$ is determined by the location-scale family.

**Generalized bound:** We extend this result to consider a more general notion of "matrix" smoothness (Thm. 5) reflecting that the sensitivity of $\nabla_{\boldsymbol{z}} \log p(\boldsymbol{z}, \boldsymbol{x})$ to changes in $\boldsymbol{z}$ may depend on the direction of change.

**Data Subsampling:** We again extend this result to consider data subsampling (Thm. 6). In particular, we observe that *non-uniform* subsampling gives tighter bounds.

In all cases, we show that the bounds are unimprovable. We experimentally compare these bounds to empirical variance.

## 2 Setup

Given some "black box" function $f$, this paper studies estimating gradients of functions $l$ of the form $l(\boldsymbol{w}) = \mathbb{E}_{\mathsf{z} \sim q_{\boldsymbol{w}}} f(\mathsf{z})$. Now, suppose some base distribution $s$ and mapping $\mathcal{T}_{\boldsymbol{w}}$ are known such that if $\mathsf{u} \sim s$, then $\mathcal{T}_{\boldsymbol{w}}(\mathsf{u}) \sim q_{\boldsymbol{w}}$. Then, $l$ can be written as

$$l(\boldsymbol{w}) = \mathbb{E}_{\mathsf{u} \sim s} f(\mathcal{T}_{\boldsymbol{w}}(\mathsf{u})).$$

If we define $\mathsf{g} = \nabla_{\boldsymbol{w}} f(\mathcal{T}_{\boldsymbol{w}}(\mathsf{u}))$, then g is an unbiased estimate of $\nabla l$, i.e. $\mathbb{E} \mathsf{g} = \nabla l(\boldsymbol{w})$. The same idea can be used when $f$ is composed as a finite sum as $f(\boldsymbol{z}) = \sum_{n=1}^{N} f_n(\boldsymbol{z})$. If $N$ is large, even evaluating $f$ once might be expensive. However, take any positive distribution $\pi$ over $n \in \{1, \cdots, N\}$ and sample $\mathsf{n} \sim \pi$ independently of u. Then, if we define $\mathsf{g} = \nabla_{\boldsymbol{w}} \pi(\mathsf{n})^{-1} f_{\mathsf{n}}(\mathcal{T}_{\boldsymbol{w}}(\mathsf{u}))$, this is again an unbiased estimator with $\mathbb{E} \mathsf{g} = \nabla l(\boldsymbol{w})$.

Convergence rates in stochastic optimization depend on the variability of the gradient estimator, typically either via the expected squared norm (ESN) $\mathbb{E} \|\mathsf{g}\|_2^2$ or the trace of the variance $\mathrm{tr}\, \mathbb{V}\, \mathsf{g}$. These are closely related, since $\mathbb{E} \|\mathsf{g}\|_2^2 = \mathrm{tr}\, \mathbb{V}\, \mathsf{g} + \|\mathbb{E} \mathsf{g}\|_2^2$.

The goal of this paper is to bound the variability of g for reparameterization / path estimators of g. This requires making assumptions about (i) the transformation function $\mathcal{T}_{\boldsymbol{w}}$ and base distribution $s$ (which determine $q_{\boldsymbol{w}}$) and (ii) the target function $f$.

Here, we are interested in the case of affine mappings. We use the mapping[17]

$$\mathcal{T}_{\boldsymbol{w}}(\boldsymbol{u}) = C\boldsymbol{u} + \boldsymbol{m},$$

where $\boldsymbol{w} = (\boldsymbol{m}, C)$ is a single vector of all parameters. This is the most common mapping used to represent location-scale families. That is, if $\mathsf{u} \sim s$ then $\mathcal{T}_{\boldsymbol{w}}(\mathsf{u})$ is equal in distribution to a location-scale family distribution. For example, if $s = \mathcal{N}(0, I)$ then $\mathcal{T}_{\boldsymbol{w}}(\mathsf{u})$ is equal in distribution to $\mathcal{N}(\boldsymbol{m}, CC^\top)$.

We will refer to the base distribution as **standardized** if the components of $\mathsf{u} = (\mathsf{u}_1, \cdots, \mathsf{u}_d) \sim s$ are iid with $\mathbb{E}\, \mathsf{u}_1 = \mathbb{E}\, \mathsf{u}_1^3 = 0$ and $\mathbb{V}\, \mathsf{u}_1 = 1$. The bounds will depend on the fourth moment $\kappa = \mathbb{E}[\mathsf{u}_1^4]$, but are otherwise independent of $s$.

To apply these estimators to VI, choose $f(\boldsymbol{z}) = \log(\boldsymbol{z}, \boldsymbol{x})$. Then $\mathrm{ELBO}(\boldsymbol{w}) = l(\boldsymbol{w}) + h(\boldsymbol{w})$ where $h$ is the entropy of $q_{\boldsymbol{w}}$. Stochastic estimates of the gradient of $l$ can be employed in a stochastic gradient method to maximize the ELBO. To model the stochastic setting, suppose that $X = (\boldsymbol{x}_1, \cdots, \boldsymbol{x}_N)$ are iid and $p(\boldsymbol{z}, X) = p(\boldsymbol{z}) \prod_{n=1}^{N} p(\boldsymbol{x}_n | \boldsymbol{z})$. Then, one may choose, e.g. $f_n(\boldsymbol{z}) =$

$\frac{1}{N} \log p(\boldsymbol{z}) + \log p(\boldsymbol{x}_n | \boldsymbol{z})$. The entropy $h$ is related to the (constant) entropy of the base distribution as $h(\boldsymbol{w}) = \mathrm{Entropy}(s) + \log |C|$.

The main bounds of this paper concern estimators for the gradient of $l(\boldsymbol{w})$ alone, disregarding $h(\boldsymbol{w})$. There are two reasons for this. First, in location-scale families, the exact gradient of $h(\boldsymbol{w})$ is known. Second, if one uses a stochastic estimator for $h(\boldsymbol{w})$, this can be "absorbed" into $l(\boldsymbol{w})$ to some degree. This is discussed further in Sec. 5.

## 3 Variance Bounds

### 3.1 Technical Lemmas

We begin with two technical lemmas which will do most of the work in the main results. Both have (somewhat laborious) proofs in Sec. 7 (Appendix). The first lemma relates the norm of the *parameter* gradient of $f(\mathcal{T}_{\boldsymbol{w}}(\boldsymbol{u}))$ (with respect to $\boldsymbol{w}$) to the norm of the gradient of $f(\boldsymbol{z})$ itself, evaluated at $\boldsymbol{z} = \mathcal{T}_{\boldsymbol{w}}(\boldsymbol{u})$.

**Lemma 1.** *For any $\boldsymbol{w}$ and $\boldsymbol{u}$, $\|\nabla_{\boldsymbol{w}} f(\mathcal{T}_{\boldsymbol{w}}(\boldsymbol{u}))\|_2^2 = \|\nabla f(\mathcal{T}_{\boldsymbol{w}}(\boldsymbol{u}))\|_2^2 \left(1 + \|\boldsymbol{u}\|_2^2\right).$*

The proof is tedious but essentially amounts to calculating the derivative with respect to each component of $\boldsymbol{w}$ (i.e. entries $\boldsymbol{m}_i$ and $C_{ij}$), summing the square of all entries, and simplifying. The second lemma gives a closed-form for the expectation of a closely related expression that will appear in the proof of Thm. 3 as a consequence of applying Lem. 1.

**Lemma 2.** *Let $\mathsf{u} \sim s$ for $s$ standardized with $\mathsf{u} \in \mathbb{R}^d$ and $\mathbb{E}_{\mathsf{u} \sim s} \mathsf{u}_i^4 = \kappa$. Then for any $\bar{\boldsymbol{z}}$,*

$$\mathbb{E} \|\mathcal{T}_{\boldsymbol{w}}(\mathsf{u}) - \bar{\boldsymbol{z}}\|_2^2 \left(1 + \|\mathsf{u}\|_2^2\right) = (d+1) \|\boldsymbol{m} - \bar{\boldsymbol{z}}\|_2^2 + (d+\kappa) \|C\|_F^2.$$

Again, the proof is tedious but based on simple ideas: Substitute the definition of $\mathcal{T}_{\boldsymbol{w}}$ into the left-hand side and expand all terms. This gives terms between zeroth and fourth order (in $\mathsf{u}$). Calculating the exact expectation of each and simplifying using the assumption that $s$ is standardized gives the result.

### 3.2 Basic Variance Bound

Given these two lemmas, we give our major technical result, bounding the variability of a reparameterization-based gradient estimator. This will be later be extended to consider data subsampling, and a generalized notion of smoothness. Note that we do *not* require that $f$ be convex.

**Theorem 3.** *Suppose $f$ is $M$-smooth, $\bar{\boldsymbol{z}}$ is a stationary point of $f$, and $s$ is standardized with $\mathsf{u} \in \mathbb{R}^d$ and $\mathbb{E} \mathsf{u}_i^4 = \kappa$. Let $\mathsf{g} = \nabla_{\boldsymbol{w}} f(\mathcal{T}_{\boldsymbol{w}}(\mathsf{u}))$ for $\mathsf{u} \sim s$. Then,*

$$\mathbb{E} \|\mathsf{g}\|_2^2 \le M^2 \left((d+1) \|\boldsymbol{m} - \bar{\boldsymbol{z}}\|_2^2 + (d+\kappa) \|C\|_F^2\right). \tag{2}$$

*Moreover, this result is unimprovable without further assumptions.*

*Proof.* We expand the definition of g, and use the above lemmas and the smoothness of $f$.

$$\begin{aligned}
\mathbb{E} \|\mathsf{g}\|_2^2 &= \mathbb{E} \|\nabla_{\boldsymbol{w}} f(\mathcal{T}_{\boldsymbol{w}}(\mathsf{u}))\|_2^2 && \text{(Definition of g)} \\
&= \mathbb{E} \|\nabla f(\mathcal{T}_{\boldsymbol{w}}(\mathsf{u}))\|_2^2 (1 + \|\mathsf{u}\|_2^2) && \text{(Lem. 1)} \\
&= \mathbb{E} \|\nabla f(\mathcal{T}_{\boldsymbol{w}}(\mathsf{u})) - \nabla f(\bar{\boldsymbol{z}})\|_2^2 (1 + \|\mathsf{u}\|_2^2) && (\nabla f(\bar{\boldsymbol{z}}) = 0) \\
&\le \mathbb{E} M^2 \|\mathcal{T}_{\boldsymbol{w}}(\mathsf{u}) - \bar{\boldsymbol{z}}\|_2^2 (1 + \|\mathsf{u}\|_2^2) && (f \text{ is smooth}) \\
&= M^2 \left((d+1) \|\boldsymbol{m} - \bar{\boldsymbol{z}}\|_2^2 + (d+\kappa) \|C\|_F^2\right). && \text{(Lem. 2)}
\end{aligned}$$

To see that this is unimprovable without further assumptions, observe that the only inequality is using the smoothness on $f$ to bound the norm of the difference of gradients at $\mathcal{T}_{\boldsymbol{w}}(u)$ and at $\bar{\boldsymbol{z}}$. But for $f(\boldsymbol{z}) = \frac{M}{2} \|\boldsymbol{z} - \bar{\boldsymbol{z}}\|_2^2$ this inequality is tight. Thus, for any $M$ and $\bar{\boldsymbol{z}}$, there is a function $f$ satisfying the assumptions of the theorem such that Eq. (2) is an equality. $\square$

With a small amount of additional looseness, we can cast Eq. (2) into a more intuitive form. Define $\bar{\boldsymbol{w}} = (\bar{\boldsymbol{z}}, 0_{d \times d})$, where $0_{d \times d}$ is a $d \times d$ matrix of zeros. Then, $\|\boldsymbol{w} - \bar{\boldsymbol{w}}\|_2^2 = \|\boldsymbol{m} - \bar{\boldsymbol{z}}\|_2^2 + \|C\|_F^2$, so we can slightly relax Eq. (2) to the more user-friendly form of

$$\mathbb{E}\|\mathsf{g}\|_2^2 \leq (d + \kappa)M^2 \|\boldsymbol{w} - \bar{\boldsymbol{w}}\|_2^2. \tag{3}$$

The only additional looseness is bounding $d + 1 \leq d + \kappa$. This is justified since when $s$ is standardized, $\kappa = \mathsf{u}_i^4$ is the kurtosis, which is at least one. Here, $\kappa$ is determined by $s$ and does not depend on the dimensionality. For example, if $s$ is Gaussian, $\kappa = 3$. Thus, Eq. (3) will typically not be much looser than Eq. (2).

Intuitively, $\bar{\boldsymbol{w}}$ are parameters that concentrate $q$ entirely at a stationary point of $f$. It is not hard to show that $\|\boldsymbol{w} - \bar{\boldsymbol{w}}\|^2 = \mathbb{E}_{\mathsf{z} \sim q_{\boldsymbol{w}}} \|\mathsf{z} - \bar{\boldsymbol{z}}\|^2$. Thus, Eq. (3) intuitively says that $\mathbb{E}\|\mathsf{g}\|^2$ is bounded in terms of how far far the average point sampled from $q_{\boldsymbol{w}}$ is from $\bar{\boldsymbol{z}}$. Since $f$ need not be convex, there might be multiple stationary points. In this case, Thm. 3 holds simultaneously for all of them.

### 3.3 Generalized Smoothness

Since the above bound is not improvable, tightening it requires stronger assumptions. The tightness of Thm. 3 is determined by the smoothness condition that the difference of gradients at two points is bounded as $\|\nabla f(y) - \nabla f(z)\|_2 \leq M \|y - z\|_2$. For some problems, $f$ may be much smoother in *certain directions* than others. In such cases, the smoothness constant $M$ will need to reflect the worst-case direction. To produce a tighter bound for such situations, we generalize the notion of smoothness to allow $M$ to be a symmetric matrix.

**Definition 4.** $f$ is $M$-**matrix-smooth** if $\|\nabla f(\boldsymbol{y}) - \nabla f(\boldsymbol{z})\|_2 \leq \|M(\boldsymbol{y} - \boldsymbol{z})\|_2$ (for symmetric $M$).

We can generalize the result in Thm. 3 to functions with this matrix-smoothness condition. The proof is very similar. The main difference is that after applying the smoothness condition, the matrix $M$ needs to be "absorbed" into the parameters $\boldsymbol{w} = (\boldsymbol{m}, C)$ before applying Lem. 2.

**Theorem 5.** *Suppose $f$ is $M$-matrix smooth, $\bar{\boldsymbol{z}}$ is a stationary point of $f$, and $s$ is standardized with $\mathsf{u} \in \mathbb{R}^d$ and $\mathbb{E}\,\mathsf{u}_i^4 = \kappa$. Let $\mathsf{g} = \nabla_{\boldsymbol{w}} f(\mathcal{T}_{\boldsymbol{w}}(\mathsf{u}))$ for $\mathsf{u} \sim s$. Then,*

$$\mathbb{E}\|\mathsf{g}\|_2^2 \leq (d + 1) \|M(\boldsymbol{m} - \bar{\boldsymbol{z}})\|_2^2 + (d + \kappa) \|MC\|_F^2. \tag{4}$$

*Moreover, this result is unimprovable without further assumptions.*

*Proof.* The proof closely mirrors that of Thm. 3. Here, given $\boldsymbol{w} = (\boldsymbol{m}, C)$, we define $\boldsymbol{v} = (M\boldsymbol{m}, MC)$, to be $\boldsymbol{w}$ with $M$ "absorbed" into the parameters.

$$\begin{aligned}
\mathbb{E}\|\mathsf{g}\|_2^2 &= \mathbb{E}\|\nabla_{\boldsymbol{w}} f(\mathcal{T}_{\boldsymbol{w}}(\mathsf{u}))\|_2^2 &&\text{Definition of }\mathsf{g}) \\
&= \mathbb{E}\|\nabla f(\mathcal{T}_{\boldsymbol{w}}(\mathsf{u}))\|_2^2 (1 + \|\mathsf{u}\|_2^2) &&\text{(Lem. 1)} \\
&= \mathbb{E}\|\nabla f(\mathcal{T}_{\boldsymbol{w}}(\mathsf{u})) - \nabla f(\bar{\boldsymbol{z}})\|_2^2 (1 + \|\mathsf{u}\|_2^2) &&(\nabla f(\bar{\boldsymbol{z}}) = 0) \\
&\leq \mathbb{E}\|M(\mathcal{T}_{\boldsymbol{w}}(\mathsf{u}) - \bar{\boldsymbol{z}})\|_2^2 (1 + \|\mathsf{u}\|_2^2) &&(f \text{ is smooth}) \\
&= \mathbb{E}\|\mathcal{T}_{\boldsymbol{v}}(\mathsf{u}) - M(\bar{\boldsymbol{z}} - \boldsymbol{m})\|_2^2 (1 + \|\mathsf{u}\|_2^2) &&\text{(Absorb }M\text{ into }\boldsymbol{v}) \\
&= (d + 1) \|M\boldsymbol{m} - M\bar{\boldsymbol{z}}\|_2^2 + (d + \kappa) \|MC\|_F^2. &&\text{(Lem. 2)}
\end{aligned}$$

To see that this is unimprovable, observe that the only inequality is the matrix-smoothness condition on $f$. But for $f(z) = \frac{1}{2}(z - \bar{z})^\top M(z - \bar{z})$, the difference of gradients $\|\nabla f(\boldsymbol{y}) - \nabla f(\boldsymbol{z})\|_2 = \|M(\boldsymbol{y} - \boldsymbol{z})\|_2$ is an equality. Thus, for any $M$ and $\bar{z}$, there is an $f$ satisfying the assumptions of the theorem such that the bound in Eq. (4) is an equality. $\square$

It's easy to see that this reduces to Thm. 3 in the case that $f$ is smooth in the standard sense– this corresponds to the situation where $M$ is some constant times the identity. Alternatively, one can simply observe that the two results are the same if $M$ is a scalar. Thus, going forward we will use Eq. (4) to represent the result with either type of smoothness assumption on $f$.

## 3.4  Subsampling

Often, the function $f(\boldsymbol{z})$ takes the form of a sum over other functions $f_n(\boldsymbol{z})$, typically representing different data. Write this as

$$f(\boldsymbol{z}) = \sum_{n=1}^{N} f_n(\boldsymbol{z}).$$

To estimate the gradient of $\mathbb{E}_{\mathsf{u} \sim s} f(\mathcal{T}_{\boldsymbol{w}}(\mathsf{u}))$, one can save time by using "subsampling": That is, draw a random $n$, and then estimate the gradient of $\mathbb{E}_{\mathsf{u} \sim s} f_n(\mathcal{T}_{\boldsymbol{w}}(\mathsf{u}))$. The following result bounds this procedure. It essentially just takes a set of estimators, one corresponding to each function $f_n$, bounds their expected squared norm using the previous theorems, and then combines these.

**Theorem 6.** *Suppose $f_n$ is $M_n$-matrix-smooth, $\bar{\boldsymbol{z}}_n$ is a stationary point of $f_n$, and $s$ is standardized with $\mathsf{u} \in \mathbb{R}^d$ and $\mathbb{E}\, \mathsf{u}_i^4 = \kappa$. Let $\mathsf{g} = \frac{1}{\pi(\mathsf{n})} \nabla f_{\mathsf{n}}(\mathcal{T}_{\boldsymbol{w}}(\mathsf{u}))$ for $\mathsf{u} \sim s$ and $\mathsf{n} \sim \pi$ independent. Then,*

$$\mathbb{E}\, \|\mathsf{g}\|_2^2 \leq \sum_{n=1}^{N} \frac{1}{\pi(n)} \left( (d+1) \|M_n(\boldsymbol{m} - \bar{\boldsymbol{z}}_n)\|_2^2 + (d+\kappa) \|M_n C\|_F^2 \right). \tag{5}$$

*Moreover, this result is unimprovable without further assumptions.*

*Proof.* Consider a simple lemma: Suppose $\mathsf{a}_1 \cdots \mathsf{a}_N$ are independent random vectors and $\pi$ is any distribution over $1 \cdots N$. Let $\mathsf{b} = \mathsf{a}_{\mathsf{n}}/\pi(\mathsf{n})$ for $\mathsf{n} \sim \pi$, where $\mathsf{n}$ is independent of $\mathsf{a}_n$. It is easy to show that $\mathbb{E}\, \mathsf{b} = \sum_{n=1}^{N} \mathbb{E}\, \mathsf{a}_n$ and $\mathbb{E}\, \|\mathsf{b}\|_2^2 = \sum_n \mathbb{E}\, \|\mathsf{a}_n\|_2^2 / \pi(n)$. The result follows from applying this with $\mathsf{a}_n = \nabla_{\boldsymbol{w}} f_n(\mathcal{T}_{\boldsymbol{w}}(\mathsf{u}))$, and then bounding $\mathbb{E}\, \|\mathsf{a}_n\|_2^2$ using Thm. 5.

Again, in this result the only source of looseness is the use of the smoothness bound for the component functions $f_n$. Accordingly, the result can be shown to be unimprovable: For any set of stationary points $\bar{\boldsymbol{z}}$ and smoothness parameters $M_n$ we can construct functions $f_n$ (as in Thm. 5) for which the previous theorems are tight and thus this result is also tight. $\qquad\square$

This result generalizes all the previous bounds: Thm. 5 is the special case when $N = 1$, while Thm. 3 is the special-case when $N = 1$ and $f_1$ is $M_1$-smooth (for a scalar $M_1$). The case where $N > 1$ but $f_n$ is $M_n$-smooth (for scalar $M_n$) is also useful– the bound in Eq. (5) remains valid, but with a scalar $M_n$.

# 4  Empirical Evaluation

## 4.1  Model and Datasets

We consider Bayesian linear regression and logistic regression models on various datasets (Table 1). Given data $\{(\boldsymbol{x}_1, y_1), \cdots (\boldsymbol{x}_N, y_N)\}$, let $\boldsymbol{y}$ be a vector of all $y_n$ and $X$ a matrix of all $\boldsymbol{x}_n$. We assume a Gaussian prior so that $p(\boldsymbol{z}, \boldsymbol{y}|X) = \mathcal{N}(\boldsymbol{z}|0, \sigma^2 I) \prod_{n=1}^{N} p(y_n|\boldsymbol{z}, \boldsymbol{x}_n)$. For linear regression, $p(y_n|\boldsymbol{z}, \boldsymbol{x}_n) = \mathcal{N}(y_n|\boldsymbol{z}^\top \boldsymbol{x}_i, \rho^2)$, while for logistic regression, $p(y_n|\boldsymbol{z}, \boldsymbol{x}_n) = \mathrm{Sigmoid}(y_n \boldsymbol{z}^\top \boldsymbol{x}_n)$. For both models we use a prior of $\sigma^2 = 1$. For linear regression, we set $\rho^2 = 4$.

To justify the use of VI, apply the decomposition in Eq. (1) substituting $p(\boldsymbol{z}, \boldsymbol{y}|X)$ in place of $p(\boldsymbol{z}, \boldsymbol{x})$ to get that

$$\log p(\boldsymbol{y}|X) = \mathop{\mathbb{E}}_{\mathsf{z} \sim q} \log \frac{p(\mathsf{z}, \boldsymbol{y}|X)}{q(\mathsf{z})} + KL\left( q(\mathsf{z}) \| p(\mathsf{z}|\boldsymbol{y}, X) \right).$$

Thus, adjusting the parameters of $q$ to maximize the first term on the right tightens a lower-bound on the conditional log likelihood $\log p(\boldsymbol{y}|X)$ and minimizes the divergence from $q$ to the posterior. So, we again take our goal as maximizing $l(\boldsymbol{w}) + h(\boldsymbol{w})$. In the batch setting, $f(\boldsymbol{z}) = \log p(\mathsf{z}, \boldsymbol{y}|X)$, while with subsampling, $f_n(\boldsymbol{z}) = \frac{1}{N} \log p(\boldsymbol{z}) + \log p(y_n|\boldsymbol{z}, \boldsymbol{x}_n)$.

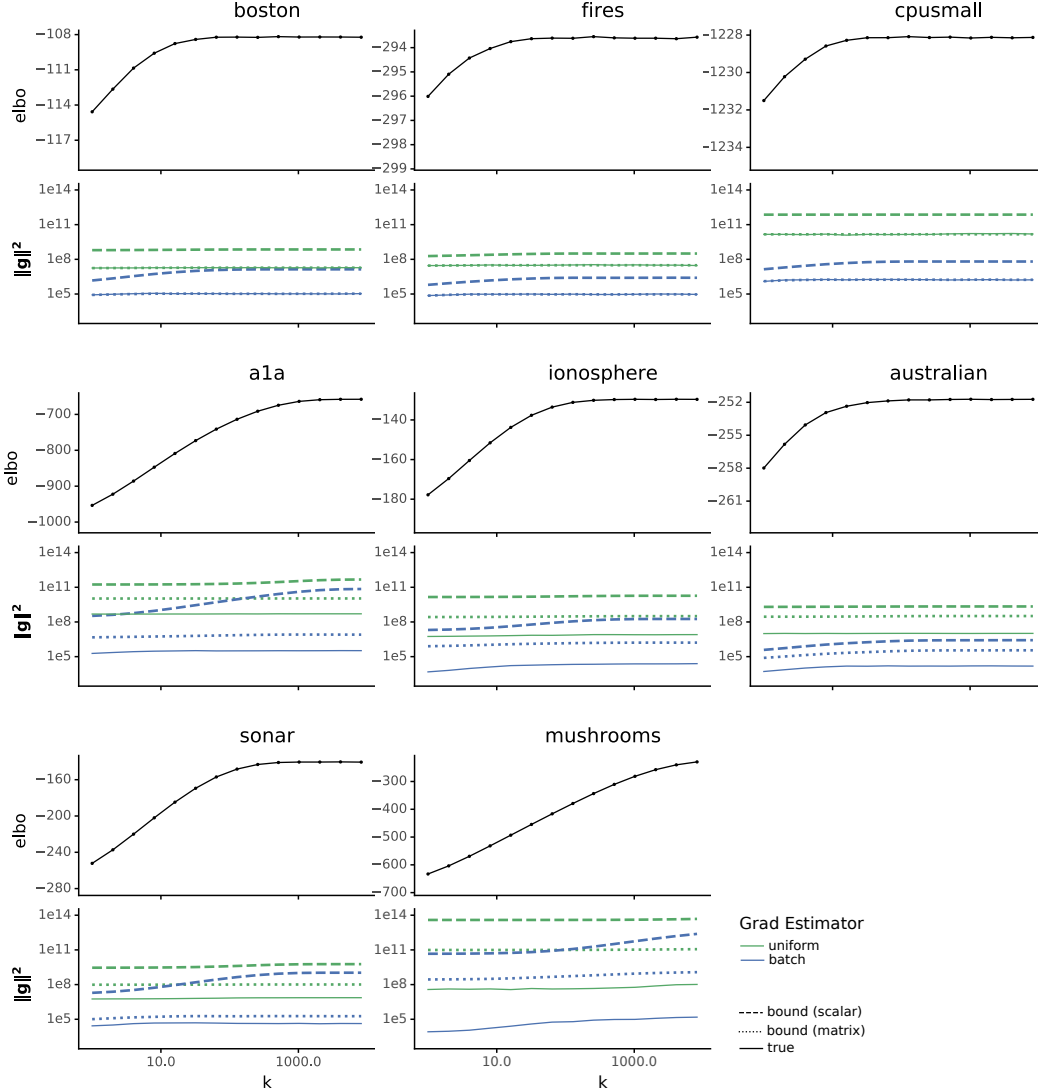

Figure 1: **How loose are the bounds compared to reality?** *Odd Rows*: Evolution of the ELBO during the single optimization trace used to compare all estimators. *Even Rows*: True and bounded variance with gradients estimated in "`batch`" (using the full dataset in each evaluation) and "`uniform`" (stochastically with $\pi(n) = 1/N$). The first two rows are for linear regression models, while the rest are for logistic regression. *Key Observations*: (i) Batch estimation is lower-variance but higher cost (ii) variance with stochastic estimation varies little over time (iii) using matrix smoothness significantly tightens bounds – and is exact for linear regression models.

Sec. 8 shows that if $0 \leq \phi''(t) \leq \theta$, then $\sum_{n=1}^{N} \phi(\boldsymbol{a}_n^\top \boldsymbol{z} + b_n)$ is $M$-matrix-smooth for $M = \theta \sum_{i=1}^{N} \boldsymbol{a}_i \boldsymbol{a}_i^\top$. Applying this[1] gives that $f(\boldsymbol{z})$ and $f_n(\boldsymbol{z})$ are matrix-smooth for

$$M = \frac{1}{\sigma^2} I + c \sum_{n=1}^{N} \boldsymbol{x}_n \boldsymbol{x}_n^\top, \text{ and } M_n = \frac{1}{N\sigma^2} I + c\, \boldsymbol{x}_n \boldsymbol{x}_n^\top,$$

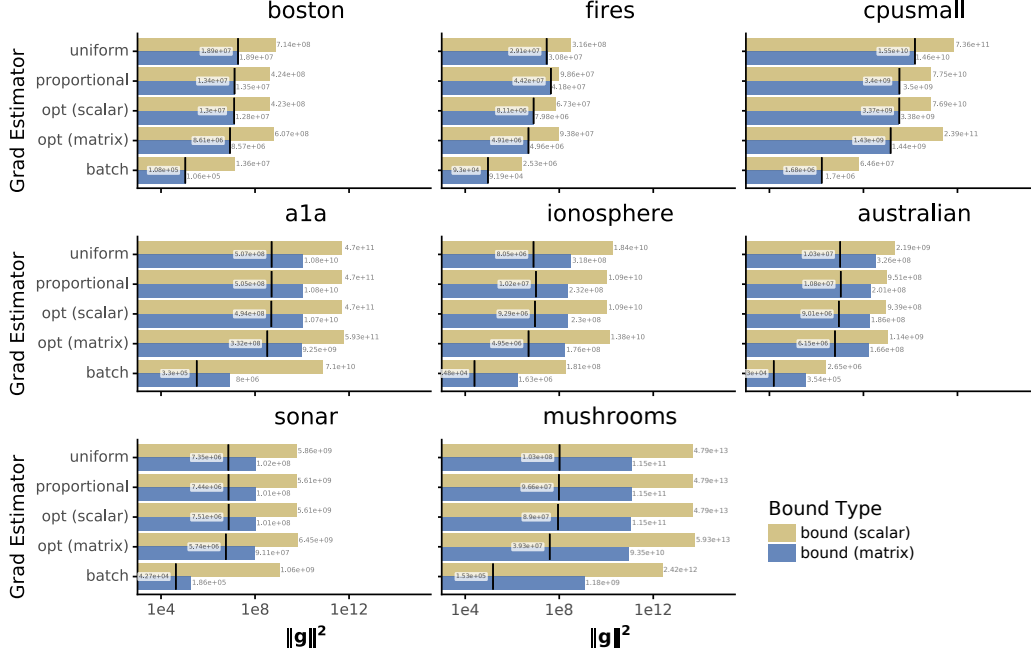

Figure 2: **Tightening variance bounds reduces true variance.** A comparison of the true (vertical bars) and bounded $\mathbb{E}\left\Vert \mathbf{g}\right\Vert^2$ values produced using five different gradient estimators. `Batch` does not use subsampling. `Uniform` uses subsampling $\pi(n) = 1/N$, `proportional` uses $\pi(n) \propto M_n$, `opt (scalar)` numerically optimizes $\pi(n)$ to tighten Eq. (5) with a scalar $M_n$ and `opt (matrix)` tightens Eq. (5) with a matrix $M_n$. For each sampling strategy, we show the variance bound both with a scalar and matrix $M_n$. Uniform sampling has true and bounded values of $\mathbb{E}\left\Vert \mathbf{g}\right\Vert^2$ ranging between 1.5x and 10x higher than those for sampling with $\pi$ numerically optimized.

where $c = 1/\rho^2$ for linear regression, and $c = 1/4$ for logistic regression. Taking the spectral norm of these matrices gives scalar smoothness constants. With subsampling, this is $\Vert M_n \Vert_2 = \frac{1}{\sigma^2 N} + c\Vert \boldsymbol{x}_n \Vert^2$.

| Dataset | Type | # data | # dims |
|---|---|---|---|
| boston | r | 506 | 13 |
| fires | r | 517 | 12 |
| cpusmall | r | 8192 | 13 |
| a1a | c | 1695 | 124 |
| ionosphere | c | 351 | 35 |
| australian | c | 690 | 15 |
| sonar | c | 208 | 61 |
| mushrooms | c | 8124 | 113 |

## 4.2 Evaluation of Bounds

To enable a clear comparison of of different estimators and bounds, we generate a single optimization trace of parameter vectors $\boldsymbol{w}$ for each dataset. All comparisons use this same trace. These use a conservative optimization method: Find a maximum $\bar{\boldsymbol{z}}$ and then initialize to $\boldsymbol{w} = (\bar{\boldsymbol{z}}, 0)$. Then, optimization

Table 1: Regression (r) and classification (c) datasets

uses proximal stochastic gradient descent (with the proximal operator reflecting $h$) with a step size of $1/M$ (the scalar smoothness constant) and 1000 evaluations for each gradient estimate.

Fig. 1 shows the evolution of the ELBO along with the variance of gradient estimation either in batch or stochastically with a uniform distribution over data. For each iteration and estimator, we plot the empirical $\Vert \mathbf{g} \Vert^2$ along with this paper's bounds using either scalar or matrix smoothness.

## 4.3 Sampling distributions

With subsampling, variability depends on the sampling distribution $\pi$. We consider uniform sampling as well as three strategies that attempt to tighten the bound in Thm. 6. In general, $\sum_n f(n)^2/\pi(n)$ is minimized over distributions $\pi$ by $\pi(n) \propto |f(n)|$. Thus, the tightest bound is given by

$$\pi^*_{\boldsymbol{w}}(n) \propto \sqrt{(d+1)\left\| M_n(\boldsymbol{m} - \bar{\boldsymbol{z}}_n) \right\|_2^2 + (d+\kappa)\left\| M_n C \right\|_F^2}. \tag{6}$$

We call this "opt (scalar)" or "opt (matrix)" when $M_n$ is a scalar or matrix, respectively. We also consider a "proportional" heuristic with $\pi(n) \propto M_n$ for a scalar $M_n$. Sampling from Eq. (6) appears to require calculating the right-hand side for each $n$ and then normalizing, which may not be practical for large datasets. While there are obvious heuristics for recursively approximating $\pi^*$ during an optimization, to maintain focus we do not pursue these ideas here.

Fig. 2 shows the empirical and true variance at the final iteration of the optimization shown in Fig. 1. The basic conclusion is that using a more careful sampling distribution reduces both true and empirical variance.

# 5 Discussion

## 5.1 Related work

Xu et al. [21] compute the variance of a reparameterization estimator applied to a quadratic function, when the variational distribution is a fully-factorized Gaussian. This paper can be seen as extending this result to more general densities (full-rank location-scale families) and more general target functions (smooth functions).

Fan et al. [4] give an abstract variance bound for RP estimators. Essentially, they argue that if $\mathsf{g}_i = \nabla_{w_i} f(\mathcal{T}_{\boldsymbol{w}}(\mathsf{u}))$ and $\nabla_{w_i} f(\mathcal{T}_{\boldsymbol{w}}(\boldsymbol{u}))$ is $M$-smooth as a function of $\boldsymbol{u}$, then $\mathbb{V}[\mathsf{g}_i] \leq M^2 \pi^2 / 4$ when $\mathsf{u} \sim \mathcal{N}(0, I)$. While this result is fairly abstract – there is no proof that the smoothness assumption holds for any particular $M$ with any particular $f$ and $\mathcal{T}_{\boldsymbol{w}}$ – it is similar in spirit to the results in this paper.

## 5.2 Variance vs Expected Squared Norms

The above results are on the the expected squared norm (ESN) of the gradient $\mathbb{E}\left\| \mathsf{g} \right\|^2$. Some stochastic gradient convergence rates instead consider (the trace of) the variance $\mathbb{V}[\mathsf{g}]$. Since $\mathrm{tr}\,\mathbb{V}[\mathsf{g}] = \mathbb{E}\left\| \mathsf{g} \right\|^2 - \left\| \mathbb{E}\,\mathsf{g} \right\|^2$, ESN bounds are valid as variance bounds. Still, one can ask if these bounds are loose. The following (proof in Sec. 7.3) gives a lower-bound that shows that there is not much to gain from a direct bound on the variance rather than just using the ESN bound from Thm. 6.

**Theorem 7.** *For any symmetric matrices $M_1, \cdots, M_N$ and vectors $\bar{\boldsymbol{z}}_1, \cdots, \bar{\boldsymbol{z}}_N$, there are functions $f_1, \cdots, f_N$ such that (1) $f_n$ is $M_n$-matrix-smooth and has a stationary point at $\bar{\boldsymbol{z}}_n$ and (2) if $s$ is standardized with $\mathsf{u} \in \mathbb{R}^d$ and $\mathbb{E}\,\mathsf{u}_i^4 = \kappa$, then for $\mathsf{g} = \frac{1}{\pi(\mathsf{n})} \nabla f_\mathsf{n}(\mathcal{T}_{\boldsymbol{w}}(\mathsf{u}))$,*

$$\mathrm{tr}\,\mathbb{V}\left\| \mathsf{g} \right\|_2^2 \geq \sum_{n=1}^N \frac{1}{\pi(n)} \left( d\left\| M_n(\boldsymbol{m} - \bar{\boldsymbol{z}}_n) \right\|_2^2 + (d+\kappa-1)\left\| M_n C \right\|_F^2 \right).$$

When $d \gg 1$ this lower-bound is very close to the upper-bound on $\mathbb{E}\left\| \mathsf{g} \right\|^2$ in Thm. 6. Thus, under this paper's assumptions, a variance bound cannot be significantly better than an ESN bound.

## 5.3 The Entropy Term

All discussion in this paper has been for gradient estimators for $l$, while the goal is of course to optimize $l + h$. For location-scale families, $h$ is known in closed-form, meaning the exact gradient – or the proximal operator for $h$ – can be computed exactly. Still, it has been observed that if $q_{\boldsymbol{w}}$ is very close to $p(\boldsymbol{z}|\boldsymbol{x})$, cancellations mean that estimating the gradient of $h + l$ might have lower variance than the gradient of $l$ alone [12].

With any variational family, it is well-known that the gradient of the entropy can be represented as $-\nabla_{\boldsymbol{w}} \mathbb{E}_{\mathsf{z} \sim q_{\boldsymbol{w}}} \log q_{\boldsymbol{v}}(\mathsf{z})|_{\boldsymbol{v}=\boldsymbol{w}}$. That is, the dependence of $\log q_{\boldsymbol{w}}$ on $\boldsymbol{w}$ can be neglected under differentiation. Thus, if one wishes to stochastically estimate the gradient of $h$, one can treat $\log q_{\boldsymbol{v}}$ in the same way as $\log p$ when calculating gradients. Then, one could apply the analysis in this paper to $f(\boldsymbol{z}) = \log p(\boldsymbol{z}, \boldsymbol{x}) - \log q_{\boldsymbol{v}}(\boldsymbol{z})$ rather than $f(\boldsymbol{z}) = \log p(\boldsymbol{z}, \boldsymbol{x})$ as done above. It is easy to imagine

situations where subtracting $\log q_v$ (or a fraction of it) from $\log p$ would change $M_n$ and $\bar{z}_n$ in such a way as to produce a tighter bound. Thus, the bounds in this paper are consistent with practices [5, 12] where using $\log q_v$ as a control variate can reduce gradient variance.

## 5.4 Smoothness and Convergence Guarantees

At a very high level, convergence rates for stochastic gradient methods require both (1) control of the variability of the gradient estimator and (2) either convexity or Lipschitz smoothness of the objective. This paper is dedicated entirely to the first goal. Independent recent work has addressed at the second issue [3]. The basic summary is that if $f(z)$ is smooth, then $l(w)$ is smooth, and similarly if $f(z)$ is strongly convex. However, full convergence guarantees for black-box VI remain an open research problem.

## 5.5 Prospects for Generalizing Bounds to Other Variational Families

The bounds given in this paper are closely tied to location-scale families: The exact form of the reparameterization function $\mathcal{T}_w$ is used in Lem. 1 and Lem. 2, which underly the main results of Thm. 3, Thm. 5, and Eq. (4). Thus, extending our proof strategy to other variational families would require deriving new results analogous to Lem. 1 and Lem. 2 for the reparameterization function $\mathcal{T}_w$ corresponding to those new variational families. Moreover, if the exact entropy is not available for a variational family, the analysis must address the variance of the entropy gradient estimator, as discussed in Sec. 5.3.

## 5.6 Limitations

This work has several limitations. First, it applies only to location-scale families, and requires that the target objective be smooth. Second, if $\log p$ is smooth, it may still be challenging in practice to establish what the smoothness constant is. Third, we observed that even with our strongest condition of matrix smoothness, the some looseness remains in the bounds with the logistic regression examples. Since the ESN bound is unimprovable, this looseness cannot be removed without using more detailed structure of the target $\log p$. It is not obvious what this structure would be, or how it would be obtained for practical black-box inference problems.

## Footnotes

[1]For linear regression, set $\phi(t) = -t^2/(2\rho^2)$, $\boldsymbol{a}_n = \boldsymbol{x}_n$ and $b_n = y_n$ and observe that $\phi'' = -1/\rho^2$. For logistic regression, set $\phi(t) = \log \text{Sigmoid}(t)$, $\boldsymbol{a}_n = y_n \boldsymbol{x}_n$ and $b_n = 0$ and observe that $\phi'' \leq 1/4$. Adding the prior and using the triangle inequality gives the result.

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
