[Supplementary Material · Variance27_appendixonly.pdf]



Figure 3: More results in the same setting as Fig. 1 (regression data)

# 6    Additional Experimental Results

Figure 4: More results in the same setting as Fig. 1 (classification data)

# 7 Proofs

## 7.1 Proof of Lem. 1

The following result is helpful for establishing Lem. 1.

**Lemma 8.** *If $\nabla_{\boldsymbol{w}} \boldsymbol{t}_{\boldsymbol{w}}(\boldsymbol{u})$ is Jacobian-transpose of $\boldsymbol{t}_{\boldsymbol{w}}(\boldsymbol{u})$ with respect to $\boldsymbol{w}$, then*

$$\nabla_{\boldsymbol{w}} \mathcal{T}_{\boldsymbol{w}}(\boldsymbol{u})^\top \nabla_{\boldsymbol{w}} \mathcal{T}_{\boldsymbol{w}}(\boldsymbol{u}) = I(1 + \|\boldsymbol{u}\|_2^2).$$

*Proof.* We use the notation $\nabla_{\boldsymbol{w}} \mathcal{T}_{\boldsymbol{w}}(\boldsymbol{u}) = \frac{d\mathcal{T}_{\boldsymbol{w}}(\boldsymbol{u})^\top}{d\boldsymbol{w}}$, meaning that $(\nabla_{\boldsymbol{w}} \mathcal{T}_{\boldsymbol{w}}(\boldsymbol{u}))_{ij} = \frac{d\mathcal{T}_{\boldsymbol{w}}(\boldsymbol{u})_j}{dw_i}$.

Each row of $\nabla_{\boldsymbol{w}} \boldsymbol{t}_{\boldsymbol{w}}(\boldsymbol{u})$ consists of the partial derivative of $\boldsymbol{t}_{\boldsymbol{w}}(\boldsymbol{u})$ with respect to one component of $\boldsymbol{w}$. Thus, the product is

$$
\begin{aligned}
(\nabla_{\boldsymbol{w}} \mathcal{T}_{\boldsymbol{w}}(\boldsymbol{u}))^\top (\nabla_{\boldsymbol{w}} \mathcal{T}_{\boldsymbol{w}}(\boldsymbol{u})) &= \sum_i \left( \frac{d}{dw_i} \mathcal{T}(\boldsymbol{u}) \right) \left( \frac{d}{dw_i} \mathcal{T}(\boldsymbol{u}) \right)^\top \\
&= \sum_i \boldsymbol{a}_i(\boldsymbol{u}) \boldsymbol{a}_i(\boldsymbol{u})^\top.
\end{aligned}
$$

We can calculate these components as

$$
\begin{aligned}
\left( \frac{d}{dm_i} \mathcal{T}_{\boldsymbol{w}}(\boldsymbol{u}) \right) \left( \frac{d}{dm_i} \mathcal{T}_{\boldsymbol{w}}(\boldsymbol{u}) \right)^\top &= \boldsymbol{e}_i \boldsymbol{e}_i^\top \\
\left( \frac{d}{dS_{ij}} \mathcal{T}_{\boldsymbol{w}}(\boldsymbol{u}) \right) \left( \frac{d}{dS_{ij}} \mathcal{T}_{\boldsymbol{w}}(\boldsymbol{u}) \right)^\top &= (u_j \boldsymbol{e}_i) (u_j \boldsymbol{e}_i)^\top \\
&= u_j^2 \boldsymbol{e}_i \boldsymbol{e}_i^\top
\end{aligned}
$$

Adding the components up, we get that

$$
\begin{aligned}
(\nabla_{\boldsymbol{w}} \mathcal{T}_{\boldsymbol{w}}(\boldsymbol{u}))^\top (\nabla_{\boldsymbol{w}} \mathcal{T}_{\boldsymbol{w}}(\boldsymbol{u})) &= \sum_i \left( \frac{d}{dm_i} \mathcal{T}_{\boldsymbol{w}}(\boldsymbol{u}) \right) \left( \frac{d}{dm_i} \mathcal{T}_{\boldsymbol{w}}(\boldsymbol{u}) \right)^\top + \sum_{i,j} \left( \frac{d}{dS_{ij}} \mathcal{T}_{\boldsymbol{w}}(\boldsymbol{u}) \right) \left( \frac{d}{dS_{ij}} \mathcal{T}_{\boldsymbol{w}}(\boldsymbol{u}) \right)^\top \\
&= \sum_i \boldsymbol{e}_i \boldsymbol{e}_i^\top + \sum_{i,j} u_j^2 \boldsymbol{e}_i \boldsymbol{e}_i^\top \\
&= I(1 + \|\boldsymbol{u}\|_2^2).
\end{aligned}
$$

The following is the main Lemma. $\qquad \square$

**Lemma 1.** *For any $\boldsymbol{w}$ and $\boldsymbol{u}$, $\|\nabla_{\boldsymbol{w}} f(\mathcal{T}_{\boldsymbol{w}}(\boldsymbol{u}))\|_2^2 = \|\nabla f(\mathcal{T}_{\boldsymbol{w}}(\boldsymbol{u}))\|_2^2 \left( 1 + \|\boldsymbol{u}\|_2^2 \right).$*

*Proof.* Using Lemma Lem. 8, we can show that

$$
\begin{aligned}
\|\nabla_{\boldsymbol{w}} f(\mathcal{T}_{\boldsymbol{w}}(\boldsymbol{u}))\|_2^2 &= \|\nabla_{\boldsymbol{w}} \mathcal{T}_{\boldsymbol{w}}(\boldsymbol{u}) \nabla f(\mathcal{T}_{\boldsymbol{w}}(\boldsymbol{u}))\|_2^2 \\
&= \nabla f(\mathcal{T}_{\boldsymbol{w}}(\boldsymbol{u}))^\top \nabla_{\boldsymbol{w}} \mathcal{T}_{\boldsymbol{w}}(\boldsymbol{u})^\top \nabla_{\boldsymbol{w}} \mathcal{T}_{\boldsymbol{w}}(\boldsymbol{u}) \nabla f(\mathcal{T}_{\boldsymbol{w}}(\boldsymbol{u})) \\
&= \nabla f(\mathcal{T}_{\boldsymbol{w}}(\boldsymbol{u}))^\top \left( I(1 + \|\boldsymbol{u}\|_2^2) \right) \nabla f(\mathcal{T}_{\boldsymbol{w}}(\boldsymbol{u})) \\
&= \|\nabla f(\mathcal{T}_{\boldsymbol{w}}(\boldsymbol{u}))\|_2^2 \left( 1 + \|\boldsymbol{u}\|_2^2 \right).
\end{aligned}
$$

$\qquad \square$

## 7.2 Proof of Lem. 2

A few distributional properties are needed before proving Lem. 2.

**Lemma 9.** *Suppose that* $u = (u_1, \cdots, u_d)$ *is random variable over* $\mathbb{R}^d$ *with zero-mean iid components. Then*

$$
\begin{aligned}
\mathbb{E}\, uu^\top &= \mathbb{E}[u_1^2]I \\
\mathbb{E}\,\|u\|_2^2 &= d\,\mathbb{E}[u_1^2] \\
\mathbb{E}\, u(1 + \|u\|_2^2) &= \mathbf{1}\ \mathbb{E}[u_1^3] \\
\mathbb{E}\, uu^\top uu^\top &= \left((d-1)\,\mathbb{E}[u_1^2]^2 + \mathbb{E}[u_1^4]\right) I.
\end{aligned}
$$

*Proof.* ($\mathbb{E}\, uu^\top$) Take any pair of indices $i$ and $j$. Then, $\left(\mathbb{E}\, uu^\top\right)_{ij} = \mathbb{E}\, u_i u_j$. If $i \neq j$ this is zero. Otherwise it is $\mathbb{E}\, u_1^2$. Thus, $\mathbb{E}\, uu^\top = \mathbb{E}[u_1^2]I$.

($\mathbb{E}\,\|u\|_2^2$) This follows from the previous result as

$$
\mathbb{E}\,\|u\|_2^2 = \mathbb{E}\,\mathrm{tr}\, uu^\top = \mathrm{tr}\,\mathbb{E}\, uu^\top = \mathrm{tr}\,\mathbb{E}[u_1^2]I = d\,\mathbb{E}[u_1^2].
$$

($\mathbb{E}\, u(1 + \|u\|_2^2)$) If $x$ and $y$ are independent, $\mathbb{E}\, xy = (\mathbb{E}\, x)(\mathbb{E}\, y)$. Thus, since the first and third moments of $u_i$ are zero,

$$
\begin{aligned}
\mathbb{E}\, u(1 + \|u\|_2^2)_i &= \mathbb{E}\, u_i \left(1 + \sum_{j=1}^d u_j^2\right) \\
&= \mathbb{E}[u_i] + \mathbb{E}[u_i^3] + \sum_{j\neq i}\mathbb{E}[u_i]\,\mathbb{E}[u_j^2] \\
&= \mathbb{E}[u_i^3].
\end{aligned}
$$

($\mathbb{E}\, uu^\top uu^\top$) It is useful to represent this term as

$$
\begin{aligned}
\left(\mathbb{E}\, uu^\top uu^\top\right)_{ij} &= \mathbb{E}\, u_i u_j \|u\|_2^2 \\
&= \mathbb{E}\, u_i u_j \sum_k u_k^2.
\end{aligned}
$$

First, suppose that $i \neq j$. Then this is

$$
\begin{aligned}
\left(\mathbb{E}\, uu^\top uu^\top\right)_{ij} &= \mathbb{E}\, u_i u_j \sum_k u_k^2 \\
&= \mathbb{E}\, u_i u_j \left(u_i^2 + u_j^2 + \sum_{k\notin\{i,j\}} u_k^2\right). \\
&= 0.
\end{aligned}
$$

This is zero since $u_i$, $u_j$ and $u_k$ are independent, and each term contains at least one of $u_i$ or $u_j$ to the first power. Since $\mathbb{E}\, u_i = 0$, the full expectation is zero.

On the other hand, suppose that $i = j$. Then this is

$$
\begin{aligned}
\left(\mathbb{E}\, uu^\top uu^\top\right)_{ii} &= \mathbb{E}\, u_i^2 \left(u_i^2 + \sum_{k\neq i} u_k^2\right) \\
&= \mathbb{E}\left(u_i^4 + u_i^2 \sum_{k\neq i} u_k^2\right) \\
&= \mathbb{E}[u_1^4] + (d-1)\,\mathbb{E}[u_1^2]^2
\end{aligned}
$$

If we put this together, we get that

$$
\mathbb{E}\, uu^\top uu^\top = \left((d-1)\,\mathbb{E}[u_1^2]^2 + \mathbb{E}[u_1^4]\right) I.
$$

$\qquad\square$

**Lemma 2.** *Let* $\mathsf{u} \sim s$ *for* $s$ *standardized with* $\mathsf{u} \in \mathbb{R}^d$ *and* $\mathbb{E}_{\mathsf{u} \sim s}\, \mathsf{u}_i^4 = \kappa$. *Then for any* $\bar{z}$,

$$\mathbb{E}\, \|\mathcal{T}_{\boldsymbol{w}}(\mathsf{u}) - \bar{z}\|_2^2 \left(1 + \|\mathsf{u}\|_2^2\right) = (d+1)\, \|\boldsymbol{m} - \bar{z}\|_2^2 + (d + \kappa)\, \|C\|_F^2.$$

*Proof.* We simply split the expectation up and calculate each part.

$$
\begin{aligned}
\mathbb{E}\, \|\mathcal{T}_{\boldsymbol{w}}(\mathsf{u}) - \bar{z}\|_2^2 \left(1 + \|\mathsf{u}\|_2^2\right) &= \mathbb{E}\, \|C\mathsf{u} + \boldsymbol{m} - \bar{z}\|_2^2 \left(1 + \|\mathsf{u}\|_2^2\right) \\
&= \mathbb{E}\left(\|C\mathsf{u}\|_2^2 + 2(\boldsymbol{m} - \bar{z})^\top C\mathsf{u} + \|\boldsymbol{m} - \bar{z}\|_2^2\right)\left(1 + \|\mathsf{u}\|_2^2\right) \\
\mathbb{E}\, \|C\mathsf{u}\|_2^2 \left(1 + \|\mathsf{u}\|_2^2\right) &= \mathbb{E}\, \|C\mathsf{u}\|_2^2 + \mathbb{E}\, \|C\mathsf{u}\|_2^2\, \|\mathsf{u}\|_2^2 \\
\mathbb{E}\, \|C\mathsf{u}\|_2^2 &= \mathbb{E}\, \mathrm{tr}\, \mathsf{u}^\top C^\top C\mathsf{u} \\
&= \mathrm{tr}\, C^\top C\, \mathbb{E}\, \mathsf{u}\mathsf{u}^\top \\
&= \mathrm{tr}\, C^\top C\, \mathbb{E}[\mathsf{u}_1^2] I \\
&= \mathbb{E}[\mathsf{u}_1^2]\, \mathrm{tr}\, C^\top C \\
\mathbb{E}\, \|C\mathsf{u}\|_2^2\, \|\mathsf{u}\|_2^2 &= \mathbb{E}\, \mathrm{tr}\, \mathsf{u}^\top C^\top C\mathsf{u}\mathsf{u}^\top \mathsf{u} \\
&= \mathrm{tr}\, C^\top C\, \mathbb{E}\, \mathsf{u}\mathsf{u}^\top \mathsf{u}\mathsf{u}^\top \\
&= \mathrm{tr}\, C^\top C\left((d-1)\, \mathbb{E}[\mathsf{u}_1^2]^2 + \mathbb{E}[\mathsf{u}_1^4]\right) I \\
&= \left((d-1)\, \mathbb{E}[\mathsf{u}_1^2]^2 + \mathbb{E}[\mathsf{u}_1^4]\right) \mathrm{tr}\, C^\top C \\
\mathbb{E}\, \|C\mathsf{u}\|_2^2 \left(1 + \|\mathsf{u}\|_2^2\right) &= \left(\mathbb{E}[\mathsf{u}_1^2] + (d-1)\, \mathbb{E}[\mathsf{u}_1^2]^2 + \mathbb{E}[\mathsf{u}_1^4]\right) \mathrm{tr}\, C^\top C \\
\mathbb{E}(\boldsymbol{m} - \bar{z})^\top C\boldsymbol{u}\left(1 + \|\mathsf{u}\|_2^2\right) &= (\boldsymbol{m} - \bar{z})^\top C\, \mathbb{E}\,\boldsymbol{u}\left(1 + \|\mathsf{u}\|_2^2\right) \\
&= (\boldsymbol{m} - \bar{z})^\top C\, \mathbf{1}\, \mathbb{E}[\mathsf{u}_1^3] \\
&= 0 \\
\mathbb{E}\, \|\boldsymbol{m} - \bar{z}\|_2^2 \left(1 + \|\mathsf{u}\|_2^2\right) &= \|\boldsymbol{m} - \bar{z}\|_2^2\, \mathbb{E}(1 + \|\mathsf{u}\|_2^2) \\
&= \|\boldsymbol{m} - \bar{z}\|_2^2 \left(1 + d\, \mathbb{E}[\mathsf{u}_1^2]\right).
\end{aligned}
$$

Adding all this up gives that

$$\mathbb{E}\, \|\mathcal{T}_{\boldsymbol{w}}(\mathsf{u}) - \bar{z}\|_2^2 \left(1 + \|\mathsf{u}\|_2^2\right) = \left(1 + d\, \mathbb{E}[\mathsf{u}_1^2]\right) \|\boldsymbol{m} - \bar{z}\|_2^2 + \left(\mathbb{E}[\mathsf{u}_1^2] + (d-1)\, \mathbb{E}[\mathsf{u}_1^2]^2 + \mathbb{E}[\mathsf{u}_1^4]\right) \|C\|_F^2.$$

In the case that the variance is one, this becomes

$$\mathbb{E}\, \|\mathcal{T}_{\boldsymbol{w}}(\mathsf{u}) - \bar{z}\|_2^2 \left(1 + \|\mathsf{u}\|_2^2\right) = (d+1)\, \|\boldsymbol{m} - \bar{z}\|_2^2 + \left(d + \mathbb{E}[\mathsf{u}_1^4]\right) \|C\|_F^2.$$

$\square$

## 7.3  Proof of Thm. 7

**Theorem 7.** *For any symmetric matrices $M_1, \cdots, M_N$ and vectors $\bar{z}_1, \cdots, \bar{z}_N$, there are functions $f_1, \cdots, f_N$ such that (1) $f_n$ is $M_n$-matrix-smooth and has a stationary point at $\bar{z}_n$ and (2) if $s$ is standardized with $\mathsf{u} \in \mathbb{R}^d$ and $\mathbb{E}\, \mathsf{u}_i^4 = \kappa$, then for $\mathsf{g} = \frac{1}{\pi(\mathsf{n})}\nabla f_{\mathsf{n}}(\mathcal{T}_{\boldsymbol{w}}(\mathsf{u}))$,*

$$\operatorname{tr} \mathbb{V}\, \|\mathsf{g}\|_2^2 \geq \sum_{n=1}^{N} \frac{1}{\pi(n)} \left( d\, \|M_n(\boldsymbol{m} - \bar{z}_n)\|_2^2 + (d + \kappa - 1)\, \|M_n C\|_F^2 \right).$$

*Proof.* First, take any matrix $M$ and vector $\bar{z}$. Define

$$f(\boldsymbol{z}) = \frac{1}{2}(\boldsymbol{z} - \bar{z})^\top M(\boldsymbol{z} - \bar{z}).$$

We can calculate that

$$
\begin{aligned}
l(\boldsymbol{w}) &= \mathop{\mathbb{E}}_{\boldsymbol{z} \sim q_w} \frac{1}{2}(\boldsymbol{z} - \bar{z})^\top M(\boldsymbol{z} - \bar{z}) \\
&= \mathop{\mathbb{E}}_{\boldsymbol{z} \sim q_w} \frac{1}{2}\boldsymbol{z}^\top M \boldsymbol{z} - \mathop{\mathbb{E}}_{\boldsymbol{z} \sim q_w} \bar{z}^\top M \boldsymbol{z} + \mathop{\mathbb{E}}_{\boldsymbol{z} \sim q_w} \frac{1}{2}\bar{z}^\top M \bar{z} \\
&= \mathop{\mathbb{E}}_{\boldsymbol{z} \sim q_w} \frac{1}{2}\operatorname{tr} M \boldsymbol{z}\boldsymbol{z}^\top - \bar{z}^\top M \boldsymbol{m} + \frac{1}{2}\bar{z}^\top M \bar{z} \\
&= \frac{1}{2}\operatorname{tr} M(\boldsymbol{m}\boldsymbol{m}^\top + CC^\top) - \bar{z}^\top M \boldsymbol{m} + \frac{1}{2}\bar{z}^\top M \bar{z} \\
&= \frac{1}{2}\boldsymbol{m}^\top M \boldsymbol{m} + \frac{1}{2}\operatorname{tr} MCC^\top - \bar{z}^\top M \boldsymbol{m} + \frac{1}{2}\bar{z}^\top M \bar{z} \\
&= \frac{1}{2}(\boldsymbol{m} - \bar{z})^\top M(\boldsymbol{m} - \bar{z}) + \frac{1}{2}\operatorname{tr} MCC^\top.
\end{aligned}
$$

Thus, we have that

$$
\begin{aligned}
\frac{dl}{d\boldsymbol{m}} &= M(\boldsymbol{m} - \bar{z}) \\
\frac{dl}{dC} &= MC
\end{aligned}
$$

If we add up components, we get that

$$\| \mathbb{E}\, \mathsf{g}\|_2^2 = \|\nabla l(\boldsymbol{w})\|_2^2 = \|M(\boldsymbol{m} - \bar{z})\|_2^2 + \|MC\|_F^2.$$

Now, given a sequence $M_1, \cdots, M_N$ and $\bar{z}_1, \cdots, \bar{z}_N$, if we choose

$$f_n(\boldsymbol{z}) = \frac{1}{2}(\boldsymbol{z} - \bar{z}_n)^\top M_n(\boldsymbol{z} - \bar{z}_n),$$

The true gradient will be

$$
\begin{aligned}
\frac{dl}{d\boldsymbol{m}} &= \sum_{n=1}^{N} M_n(\boldsymbol{m} - \bar{z}_n) \\
\frac{dl}{dC} &= M_n C,
\end{aligned}
$$

and so, applying Jensen's inequality,

$$
\begin{aligned}
\|\mathbb{E}\,\mathbf{g}\|_2^2 &= \|\nabla l(\boldsymbol{w})\|_2^2 \\
&= \left\|\sum_{n=1}^{N} M_n(\boldsymbol{m} - \bar{\boldsymbol{z}}_n)\right\|_2^2 + \left\|\sum_{n=1}^{N} M_n C\right\|_F^2 \\
&= \left\|\sum_{n=1}^{N} \frac{1}{\pi(n)}\pi(n) M_n(\boldsymbol{m} - \bar{\boldsymbol{z}}_n)\right\|_2^2 + \left\|\sum_{n=1}^{N} \frac{1}{\pi(n)}\pi(n) M_n C\right\|_F^2 \\
&\leq \sum_{n=1}^{N} \pi(n)\left\|\frac{1}{\pi(n)} M_n(\boldsymbol{m} - \bar{\boldsymbol{z}}_n)\right\|_2^2 + \sum_{n=1}^{N} \pi(n)\left\|\frac{1}{\pi(n)} M_n C\right\|_F^2 \\
&= \sum_{n=1}^{N} \frac{1}{\pi(n)}\left(\|M_n(\boldsymbol{m} - \bar{\boldsymbol{z}}_n)\|_2^2 + \|M_n C\|_F^2\right).
\end{aligned}
$$

Thm. 6 tells us that

$$
\mathbb{E}\,\|\mathbf{g}\|_2^2 = \sum_{n=1}^{N} \frac{1}{\pi(n)}\left((d+1)\,\|M_n(\boldsymbol{m} - \bar{\boldsymbol{z}}_n)\|_2^2 + (d+\kappa)\,\|M_n C\|_F^2\right).
$$

Thus, we have that

$$
\begin{aligned}
\operatorname{tr}\mathbb{V}\,\|\mathbf{g}\|_2^2 &= \mathbb{E}\,\|\mathbf{g}\|^2 - \|\mathbb{E}\,\mathbf{g}\|^2 \\
&\geq \sum_{n=1}^{N} \frac{1}{\pi(n)}\left(d\,\|M_n(\boldsymbol{m} - \bar{\boldsymbol{z}}_n)\|_2^2 + (d+\kappa-1)\,\|M_n C\|_F^2\right).
\end{aligned}
$$

$\square$

# 8 Smoothness conditions for linear models

**Lemma 10.** *Suppose that* $f(z) = \phi(a^\top z)$, *and that* $|\phi''(t)| \leq \theta$ *for all t. Then,*

$$\|\nabla f(y) - \nabla f(z)\|_2 \leq \theta \, \|a\|_2 \, \left|a^\top(y-z)\right|.$$

*Proof.* Then, we have that

$$
\begin{aligned}
\|\nabla f(y) - \nabla f(z)\|_2 &= \left\|a\phi'(a^\top y) - a\phi'(a^\top z)\right\|_2 \\
&= \|a\|_2 \left|\phi'(a^\top y) - \phi'(a^\top z)\right| \\
&= \|a\|_2 \left|\int_{a^\top z}^{a^\top y} \phi''(t)dt\right| \\
&\leq \theta \, \|a\|_2 \left|a^\top(y-z)\right|.
\end{aligned}
$$

$\square$

**Lemma 11.** *Suppose that* $f(z) = f_0(z) + \phi(a^\top z)$ *and that* $f_0(z)$ *is* $M_0$ *smooth. Then, we have that*

$$\|\nabla f(y) - \nabla f(z)\|_2 \quad = \quad M_0 \, \|y-z\|_2 + \theta \, \|a\|_2 \left|a^\top(y-z)\right|.$$

**Lemma 12.** *Suppose that* $f(z) = \sum_{i=1}^N \phi(a_i^\top z)$ *and that* $0 \leq \phi''(t) \leq \theta$ *for all t. Then,*

$$
\begin{aligned}
\|\nabla f(y) - \nabla f(z)\|_2 &\leq \|M(y-z)\|_2 \\
M &= \theta \sum_{i=1}^N a_i a_i^\top
\end{aligned}
$$

*Proof.*

$$
\begin{aligned}
\|\nabla f(y) - \nabla f(z)\|_2 &= \left\|\sum_{i=1}^N a_i \phi'(a_i y) - \sum_{i=1}^N a_i \phi'(a_i z)\right\|_2 \\
&= \left\|\sum_{i=1}^N a_i \left(\phi'(a_i y) - \phi'(a_i z)\right)\right\|_2 \\
&= \left\|\sum_{i=1}^N a_i \int_{a_i^\top z}^{a_i^\top y} \phi''(t)dt\right\|_2 \\
&= \left\|\sum_{i=1}^N a_i \left(a_i^\top y - a_i^\top z\right) b_i\right\|_2 \\
&\quad -\theta \leq b_i \leq \theta \\
&= \left\|\sum_{i=1}^N b_i a_i a_i^\top (y-z)\right\|_2 \\
&\leq \theta \left\|\left(\sum_{i=1}^N a_i a_i^\top\right)(y-z)\right\|_2
\end{aligned}
$$

The final inequality is justified by the following claim: $\left\|\sum_{i=1}^{N} b_i a_i a_i^\top (y - z)\right\|_2^2$ is maximized over vectors $b$ with $0 \leq b_i \leq \theta$ by setting $b_i = \theta$ always. To establish this claim observe that

$$
\begin{aligned}
\frac{d}{db_k} \left\|\sum_{i=1}^{N} b_i a_i a_i^\top (y - z)\right\|_2^2 &= \frac{d}{db_k} \left(\sum_{i=1}^{N} b_i a_i a_i^\top (y - z)\right)^\top \left(\sum_{j=1}^{N} b_j a_j a_j^\top (y - z)\right) \\
&= \frac{d}{db_k} \sum_{i=1}^{N} \sum_{j=1}^{N} b_i b_j (y - z)^\top \left(a_i a_i^\top a_j a_j^\top\right)(y - z) \\
&= \frac{d}{db_k} 2 \sum_{j \neq k}^{N} b_k b_j (y - z)^\top \left(a_k a_k^\top a_j a_j^\top\right)(y - z) \\
&\quad + \frac{d}{db_k} b_k^2 (y - z)^\top \left(a_k a_k^\top a_k a_k^\top\right)(y - z) \\
&= 2 \sum_{j \neq k}^{N} b_j (y - z)^\top \left(a_k a_k^\top a_j a_j^\top\right)(y - z) \\
&\quad + 2 b_k (y - z)^\top \left(a_k a_k^\top a_k a_k^\top\right)(y - z) \\
&= 2 \sum_{j=1}^{N} b_j (y - z)^\top \left(a_k a_k^\top a_j a_j^\top\right)(y - z) \\
&= 2 \sum_{j=1}^{N} b_j \operatorname{tr}(y - z)^\top \left(a_k a_k^\top a_j a_j^\top\right)(y - z) \\
&= 2 \operatorname{tr} a_k a_k^\top \left(\sum_{j=1}^{N} b_j a_j a_j^\top\right)(y - z)(y - z)^\top \\
&= 2 a_k^\top \left(\sum_{j=1}^{N} b_j a_j a_j^\top\right)(y - z)(y - z)^\top a_k
\end{aligned}
$$

Now, both $\left(\sum_{j=1}^{N} b_j a_j a_j^\top\right)$ and $(y - z)(y - z)^\top$ are real symmetric positive definite matrices. Thus, their product has real non-negative eigenvalues. This means that

$$
\frac{d}{db_k} \left\|\sum_{i=1}^{N} b_i a_i a_i^\top (y - z)\right\|_2^2 \geq 0,
$$

i.e. the maximizing $b$ will set all entries to $\theta$. $\qquad\square$

**Theorem 13.** *Suppose that* $f(z) = \frac{c}{2} \|z\|_2^2 + \sum_{i=1}^{N} \phi(a_i^\top z)$ *and that* $0 \leq \phi''(t) \leq \theta$. *Then,*

$$
\begin{aligned}
\|\nabla f(y) - \nabla f(z)\|_2 &\leq \|M(y - z)\|_2 \\
M &= cI + \theta \sum_{i=1}^{N} a_i a_i^\top
\end{aligned}
$$

*Proof.* Suppose that $\nabla f_0(y) - \nabla f_0(z) = c(y - z)$. Then, we have that

$$
\begin{aligned}
\|\nabla f(y) - \nabla f(z)\|_2 &= \left\| \sum_{i=1}^{N} a_i \phi'(a_i y) - \sum_{i=1}^{N} a_i \phi'(a_i z) + c(y - z) \right\|_2 \\
&= \left\| \sum_{i=1}^{N} a_i \left( \phi'(a_i y) - \phi'(a_i z) \right) + c(y - z) \right\|_2 \\
&= \left\| \sum_{i=1}^{N} a_i \int_{a_i^\top z}^{a_i^\top y} \phi''(t) dt + c(y - z) \right\|_2 \\
&= \left\| \sum_{i=1}^{N} a_i \left( a_i^\top y - a_i^\top z \right) b_i + c(y - z) \right\|_2 \\
&\qquad -\theta \le b_i \le \theta \\
&= \left\| \left( cI + \sum_{i=1}^{N} b_i a_i a_i^\top \right) (y - z) \right\|_2 \\
&\le \left\| \left( cI + \theta \sum_{i=1}^{N} a_i a_i^\top \right) (y - z) \right\|_2 .
\end{aligned}
$$

The final inequality is justified by the following claim: $\left\| \sum_{i=1}^{N} b_i a_i a_i^\top (y - z) \right\|_2^2$ is maximized over vectors $b$ with $0 \le b_i \le \theta$ by setting $b_i = \theta$ always. To establish this claim observe that

$$
\begin{aligned}
\frac{d}{db_k} \left\| \left( cI + \sum_{i=1}^{N} b_i a_i a_i^\top \right) (y - z) \right\|_2^2 &= \frac{d}{db_k} \left( \left( \left( cI + \sum_{i=1}^{N} b_i a_i a_i^\top \right) (y - z) \right)^\top \left( \left( cI + \sum_{j=1}^{N} b_j a_j a_j^\top \right) (y - z) \right) \right) \\
&= 2 \left( \left( cI + \sum_{i=1}^{N} b_i a_i a_i^\top \right) (y - z) \right)^\top \frac{d}{db_k} \left( cI + \sum_{i=1}^{N} b_i a_i a_i^\top \right) (y - z) \\
&= 2 (y - z)^\top \left( cI + \sum_{i=1}^{N} b_i a_i a_i^\top \right) \left( cI + b_k a_k a_k^\top \right) (y - z) \\
&= 2 \operatorname{tr} \left( cI + \sum_{i=1}^{N} b_i a_i a_i^\top \right) \left( cI + b_k a_k a_k^\top \right) (y - z)(y - z)^\top \\
&= 2 \operatorname{tr} \left( cI + \sum_{i=1}^{N} b_i a_i a_i^\top \right) b_k a_k a_k^\top (y - z)(y - z)^\top \\
&\qquad + 2c \operatorname{tr} \left( cI + \sum_{i=1}^{N} b_i a_i a_i^\top \right) (y - z)(y - z)^\top \\
&= 2 b_k \operatorname{tr} a_k^\top (y - z)(y - z)^\top \left( cI + \sum_{i=1}^{N} b_i a_i a_i^\top \right) a_k \\
&\qquad + 2c \operatorname{tr} (y - z)^\top \left( cI + \sum_{i=1}^{N} b_i a_i a_i^\top \right) (y - z) \\
&\ge 0.
\end{aligned}
$$

The last inequality follows from the fact that

$$
\left( cI + \sum_{i=1}^{N} b_i a_i a_i^\top \right)
$$

and

$$(y - z)(y - z)^\top$$

are both real, symmetric positive definite matrices. □

# 9  Specific Models

## 9.1  Linear Model

Suppose that $p(\boldsymbol{z}) = \mathcal{N}(\boldsymbol{z}|0, \frac{1}{c}I)$ and $p(y_i|\boldsymbol{x}_i, \boldsymbol{z}) = \mathcal{N}(y_i|\boldsymbol{z}^\top \boldsymbol{x}_i, \frac{1}{b})$. Then, we have that

$$
\begin{aligned}
p(\boldsymbol{z}) \prod_i p(y_i|\boldsymbol{x}_i, \boldsymbol{z}) \quad &\propto \quad \exp\left(-\frac{1}{2c}\|\boldsymbol{z}\|^2 - \sum_i \frac{1}{2b}(y_i - \boldsymbol{z}^\top \boldsymbol{x}_i)^2\right) \\
&= \quad \exp\left(-\frac{c}{2}\|\boldsymbol{z}\|^2 - \sum_i \frac{b}{2}(y_i - \boldsymbol{z}^\top \boldsymbol{x}_i)^2\right) \\
&= \quad \exp\left(-\frac{c}{2}\|\boldsymbol{z}\|^2 - \frac{b}{2}\|\boldsymbol{y} - X\boldsymbol{z}\|_2^2\right) \\
&= \quad \exp\left(-\frac{c}{2}\|\boldsymbol{z}\|^2 - \frac{b}{2}\|\boldsymbol{y}\|_2^2 + b\boldsymbol{y}^\top X\boldsymbol{z} - \frac{b}{2}\boldsymbol{z}^\top X^\top X\boldsymbol{z}\right) \\
&\propto \quad \exp\left(b\boldsymbol{y}^\top X\boldsymbol{z} - \frac{1}{2}\boldsymbol{z}^\top \left(bX^\top X + cI\right)\boldsymbol{z}\right) \\
&= \quad \exp\left(\boldsymbol{a}^\top \boldsymbol{z} - \frac{1}{2}\boldsymbol{z}^\top \Sigma^{-1} \boldsymbol{z}\right) \\
&\propto \quad \exp\left(-\frac{1}{2}(\boldsymbol{z} - \Sigma\boldsymbol{a})\Sigma^{-1}(\boldsymbol{z} - \Sigma\boldsymbol{a})\right) \\
&= \quad \exp\left(-\frac{1}{2}(\boldsymbol{z} - \mu)\Sigma^{-1}(\boldsymbol{z} - \mu)\right)
\end{aligned}
$$

$$
\begin{aligned}
\Sigma &= \left(bX^\top X + cI\right)^{-1} \\
\mu &= \Sigma\boldsymbol{a} \\
&= \left(bX^\top X + cI\right)^{-1} bX^\top \boldsymbol{y} \\
&= \left(X^\top X + \frac{c}{b}I\right)^{-1} X^\top \boldsymbol{y}
\end{aligned}
$$

# 10  Reparameterization Stuff

## 10.1  Motivation

Suppose that $\log p(z, x)$ is something of the form

$$
\log p(z, x) = \mathbf{1}^\top \phi(Xz).
$$

We have that

$$
\nabla_z \log p(z, x) = X^\top \phi'(Xz)
$$

and that

$$
\nabla_z^2 \log p(z, x) = X^\top \phi''(Xz)X.
$$

If we suppose that $0 \le \phi'' \le \theta$ (for example this is true with logistic regression with $\theta = \frac{1}{4}$) then we have that

$$
0 \preceq \nabla_z^2 \log p(z, x) \preceq \theta X^\top X.
$$

If we were to add a uniform prior, we'd have something like

$$
cI \preceq \nabla_z^2 \log p(z, x) \preceq cI + \theta X^\top X.
$$

On the other hand, for Bayesian regression, we'd have something like

$$
\theta X^\top X \preceq \nabla_z^2 \log p(z, x) \preceq \theta X^\top X
$$

with $\theta = 1$. This offers much stronger possibilities for rescaling.

## 10.2 Divergence

Suppose that $\log p(z, x)$ is some distribution that is "poorly scaled". That is, if we compute the condition number, it is quite poor. On the other hand, it could be that for some $A$ and $b$, $\log p(Az+b, x)$ is much better-conditioned. The following lemma shows that we are free to re-scale $p$ in whatever way we want and then have $q$ target that rescaled distribution. Once that's done, we can then transform $q$ back to the original space.

**Lemma 14.** *Suppose that $p_z(z)$ is some distribution and $p_y(y)$ is the distribution of $Az + b$, $z \sim p_z$, namely*

$$p_y(y) = \frac{1}{|A|} p_z(A^{-1}(y-b)).$$

*Suppose that $q_y$ is some distribution which is "close" to $p_y$. If we define*

$$q_z(z) = |A| \, q_y(Az+b),$$

*then $KL\left(q_z \| p_z\right) = KL\left(q_y \| p_y\right).$*

## 10.3 Concrete

**Lemma 15.** *If $B \preceq C$ then $A^\top BA \preceq A^\top CA$.*

## 10.4 Proofs

**Lemma 14.** *Suppose that $p_z(z)$ is some distribution and $p_y(y)$ is the distribution of $Az + b$, $z \sim p_z$, namely*

$$p_y(y) = \frac{1}{|A|} p_z(A^{-1}(y-b)).$$

*Suppose that $q_y$ is some distribution which is "close" to $p_y$. If we define*

$$q_z(z) = |A| \, q_y(Az+b),$$

*then $KL\left(q_z \| p_z\right) = KL\left(q_y \| p_y\right).$*

*Proof.* In more detail, we know that if $y = T(z)$ then $\mathbb{P}(z = z) = \mathbb{P}(y = T(z)) \, |T'(z)|$. In our case, we use $T(z) = Az + b$ so we have that

$$p_z(z, x) = p_y(Az + b, x) \, |A|$$

Intuitively, we should correspondingly define

$$q_z(z) = q_y(Az + b, x) \, |A|.$$

Then, we have that

$$
\begin{aligned}
\mathop{\mathbb{E}}_{z \sim q_z} \log \frac{p_z(z, x)}{q_z(z, x)} &= \mathop{\mathbb{E}}_{z \sim q_z} \log \frac{p_y(Az + b, x) \, |A|}{q_y(Az + b, x) \, |A|} \\
&= \int q_z(z) \log \frac{p_y(Az + b, x)}{q_y(Az + b, x)} dz \\
&= \int |A| \, q_y(Az + b, x) \log \frac{p_y(Az + b, x)}{q_y(Az + b, x)} dz \\
&= \int q_y(y, x) \log \frac{p_y(y, x)}{q_y(y, x)} dy
\end{aligned}
$$

Where in the last line we apply

$$\int f(y) dy = \int f\left(T(z)\right) |\nabla T(z)| \, dz$$

with $f(y) = q_y(y, x) \log \frac{p_y(y,x)}{q_y(y,x)}$ and $T(z) = Az + b$. $\qquad \square$

**Lemma 16.** *If $B \preceq C$ then $A^\top BA \preceq A^\top CA$.*

*Proof.* Suppose that $B \preceq C$ meaning that $C - B$ is positive definite. Then note that

$$A^\top CA - A^\top BA = A^\top (C - B)A$$

is also positive definite, since for any $x$,

$$
\begin{aligned}
x^\top A^\top (C - B)Ax &= z^\top (C - B)z, \quad z = Ax. \\
&\geq 0.
\end{aligned}
$$

Thus we have that

$$A^\top BA \preceq A^\top CA.$$

$\square$

## 11 Gradient Variance with a Full-Covariance Quadratic

Suppose that $f(z) = \frac{1}{2}(z - \bar{z})^\top M(z - \bar{z})$. What is the gradient variance? The gradient is $\nabla f(z) = M(z - \bar{z})$. Thus, we seem to get that

$$
\begin{aligned}
\mathbb{E}_{u \sim s} \|\nabla_w f(\mathcal{T}_w(u))\|_2^2 &= \mathbb{E} \|\nabla f(\mathcal{T}_w(u))\|_2^2 \left(1 + \|u\|_2^2\right) \\
&= \mathbb{E} \|M(\mathcal{T}_w(u) - \bar{z})\|_2^2 \left(1 + \|u\|_2^2\right) \\
&= \mathbb{E} \|MCu + m - M\bar{z}\|_2^2 \left(1 + \|u\|_2^2\right) \\
&= (d + 1) \|m - M\bar{z}\|_2^2 + \left(d + \mathbb{E}[u_1^4]\right) \|MC\|_F^2 .
\end{aligned}
$$

The key thing, for this to work is showing that

$$\|\nabla f(y) - \nabla f(z)\|_2 \leq \|M(y - z)\|_2 .$$

Certainly, if we had a property like that, we would be in business.

Claim: If $f$ is $M$-smooth in the above sense, then $\frac{1}{2}z^\top Mz - f(z)$ is convex.

What does the above say about the Hessian? For very close $y$ and $z$,

$$\nabla f(y) - \nabla f(z) \approx \nabla^2 f(z)(y - z).$$

Thus the bound sort of says that

$$\left\|\nabla^2 f(z)(y - z)\right\|_2^2 \leq \|M(y - z)\|_2^2 .$$

Or, essentially, that

$$x^\top \left(\nabla^2 f(z)\right)^2 x \leq x^\top M^2 x.$$