[Reviews · NeurIPS 2019]

Reviewer 1



Strengths: The paper is clearly written and easy to follow. The author established several unimprovable variance bounds under specific assumptions. The property is appealing, though the smooth constants for most functional families are difficult to compute. The subsampling trick may be useful or be able to generalize to infinite some of functions. Weaknesses: 1. The problem discussed in this paper is relatively straightforward, even it is targeted for black box variational inference. The overall discussed topic is still relying on the existence of affine mapping for location scale distributed latent variables. The more general reparameterization trick has been discussed in [12] and "Implicit reparameterization gradients (NIPS 2018)". I admit there is no error in this manuscript, but I think that a thorough analysis for more challenging task would meet the acceptance bar of NIPS venue. 2. In the experimental section, only two toy models (Baysian linear and logistic regression) are considered. I understand the experiments are supportive to verify the variance bound proposed in this paper. However, in order to recognize the value of this work, I would like to see at least one additional experiment with amortized variational inference, such as a simple VAE whose bound may be easy to calculate. Typos: In Appendix: Eq below Line 298: \epsilon -> u Eq below Line 301: t -> T

Reviewer 2



I have only positive remarks on this paper. Originality: This paper extends theoretical results from [Xu et al] and [Fan et al] on bounds of the variance of reparameterization estimators. Quality: The paper is of high quality: - Theoretical results of high quality: assumptions precisely stated, proofs given in their gist with details in supp materials. The limitations of the results are clearly stated as well. - Experimentations are rigorous and intuitive. The results on the different types of subsampling, to optimize the variance of the estimator, are interesting. Clarity: This paper is very well written, clear in its theoretical and detailed and intuitive in its experimental part. Significance: The results are significant and of interest to the NeurIPS community.

Reviewer 3



Originality: The variance bound for the gradient estimator is novel and understanding the technical condition would be helpful in analyzing the stochastic gradient descent method. Quality: The technical results are sound and clearly highlights the novel contribution. However, it was not entirely clear how the choice of constant M or M_n would be made? Clarity: The theorems and the lemmas are clearly written. The main contribution of the paper is clearly described through the technical results. Significance: How the results on the bound on the variance of the gradient estimator is overall makes an impact in variational inference? Surely, there is the other term the entropy whose gradient will also play a part in optimizing the ELBO (w) function and hence variance of the gradient of h(w) term should also be controlled - isn't that so or am I making a mistake?

[Author Response · NeurIPS 2019]

**[R1/R3] Why affine mappings / location-scale families.** We certainly agree that it would be great to have a similar analysis for more general distributions (and we discussed this in Sec. 5.4) However, we would like to emphasize three points:

- The current analysis represents an important first step. The results of this paper are by far the strongest gradient variance bounds that have been shown to date. This paper shows one successful "path" for unimprovable bounds for location-scale families. This is an important first step for addressing more complex distributions.

- The current paper's analysis is rather difficult. The paper itself are supported by more than 8 pages of proofs in the appendix. It would be unreasonable to add additional content. We have made *every effort* to simplify the presentation and make the results here accessible, which may conceal the technical difficulty to some degree. Still, no reasonable person could call these results "straightforward" and we kindly ask that this be reconsidered after consulting the full proofs.

- Location-scale families are important! For example, the ADVI implementation in Stan is widely used and based on Gaussians (a subset of location-scale families). The Bayesian CLT means that many posteriors really are "nearly" Gaussian. The complexity of the variational distribution represents a kind of complexity/reliability/accuracy tradeoff. It remains firmly within the interest of NeurIPS to investigate Gaussian variational distributions. These are almost certainly the single-most import variational family and (before now) not much was understood about the resulting variance. This is doubly true when the analysis strategy may lead to progress on more complex variational distributions.

**[R3] clarifying whether the variance of the gradient estimator of h(w) plays a role in controlling the overall variance?** We address this in Section 5.3, though more elaboration may be helpful. Fortunately, there is no need for concern. Firstly, with location-scale families one can compute $h(w)$ (or its gradient) exactly. As we discuss in Sec. 5.2 some SGD convergence bounds can be written in terms of the gradient *variance* rather than $\mathbb{E}\|g\|^2$. Since (1) an exact entropy gradient does not increase the variance, and (2) the variance of an estimator of $\nabla l(w)$ cannot be much lower than the mean squared norm of an estimator of $\nabla l(w)$ (Sec 5.2)), the paper focuses on this latter task.

On the other hand, if the entropy will be estimated, then $\log q$ can be "absorbed" into $f$ – see the discussion on lines 230-235.

**[R1 / R3] How to choose the smoothness constant?** This indeed a limitation (as we mention in Sec. 5.4). However, keep in mind that the *vast majority* of non-stochastic optimization rates also require smoothness. Because smoothness is so widely used, many ideas have been proposed in the optimization literature for explicitly estimating the constant, e.g.

- Stochastic First- and Zeroth-Order Methods for Non-convex Stochastic Programming, Ghadimi and Lan, SIAM Journal on Optimization, 2013.

- Lipschitz gradients for global optimization in a one-point-based partitioning scheme, Kvasov and Sergeyev, Journal of Computational and Applied Mathematics, 2012.

The smoothness constant (and gradient variance) influence the convergence rate via the step-size. In practice, of course, people often manually experiment with different step-sizes. So, roughly speaking, this paper says that if one is able to tinker with step-sizes to to find $z^* = \arg\max_z p(z, x)$ then one should also be able to do VI.

**[R3] considering there is now result on controlling the variance of the gradient estimator of the VI objective, is it possible to provide a confidence interval (approximate) of the gradient estimate?.** While we aren't quite sure of the motivation, this appears possible. From the multivariate Chernoff bound, we know that if g has mean $\mu$ and a covariance matrix with singular values $\sigma = (\sigma_1 \cdots \sigma_n)$, then $\mathbb{P}[\|g - \mu\|_2 \geq k \|\sigma\|_2] \leq \frac{1}{k^2}$. So, if we know that $\mathbb{E}\|g\|_2^2 \leq c$ then $\|\sigma\|_2^2 = \sum_{i=1}^n \sigma_n^2 = \operatorname{tr} \mathbb{V}[g] \leq \mathbb{E}\|g\|_2^2 \leq c$ and so $\mathbb{P}[\|g - \mu\|_2 \geq k\sqrt{c}] \leq \frac{1}{k^2}$. Choosing a given confidence level and inverting this equation will give a confidence set for $\mu$. (A confidence set rather than interval since g is a vector.)

**[R1] I am wondering whether the author can use the functional analysis tool to approximate an arbitrary function with a representation of infinite sum of basic functions, for example, satisfying the conditional \sum M_i < \infty.** This is an interesting idea, but it's not straightforward since the sum must be over the individual sampled functions. If an arbitrary function is represented as an infinite sum of simple functions, the bound wouldn't *immediately* apply unless one could sample a simple function. Of course, with further work something along these lines might work, but that would be a paper of its one. Of course (probably unsurprisingly) we see the fact that this paper suggests directions like this as further evidence of its value.

[Meta-Review · NeurIPS 2019]

The paper provides a clear presentation of variance bounds for reparameterization gradients for location scale families. The drawback is the limited applicability of the approach. In a revision, the paper should address the following concerns: - A more detailed discussion of the role of the entropy term - A paragraph on smoothness constants - A discussion on if the proof techniques would generalize to more general classes of variational approximation. If they don't, detailing the technical hurdle that limits this line of analysis would be worthwhile